# Structure and chemical bonding in high-pressure potassium silver alloys
Nnanna Ukoji[1], Danny Rodriguez [1], Huiyao Kuang[1], Serge Desgreniers [2] & John S. Tse [1] ✉

The high-pressure structures of K-Ag alloys were examples of pressure-induced electron transfer from the electropositive potassium to the electronegative silver. We re-examined the crystal and electronic structures of $KAg_2$, $K_2Ag$, and $K_3Ag$ using powder X-ray diffraction and theoretical calculations. Our findings establish a connection between the morphologies of these three phases and the precursor face-centered cubic Ag. For $K_2Ag$, we discovered a disordered structure that better matches the X-ray pattern. Valence electron density distributions obtained from the maximum entropy method, along with charge density calculations, provide a comprehensive understanding of the evolution of chemical bonding in these systems. It was found that K atoms share their valence electrons during alloy formation, contributing to K-Ag and Ag-Ag bonds in $K_2Ag$ and $KAg_2$, while no Ag-Ag bonds are present in $K_3Ag$. These results indicate the Zintl-Klemm model may be too simplistic to describe the structure and bonding in high-pressure binary intermetallic compounds.

Discovered in the late 1990s, K-Ag alloys were the first examples of potassium forming intermetallic compounds with silver at relatively low pressure[1,2], as K and Ag do not form compounds at ambient conditions. The fast kinetics of the solid-state reactions at higher pressure was attributed to the large electronegativity difference between the two elements facilitating a charge transfer from K to Ag and forming compounds like $K_2Ag$ and $K_3Ag$[1]. This description is similar to the formation of Zintl phases[3]. Early theoretical calculations found that the band structures of $K_2Ag$ and $K_3Ag$ are remarkably similar to the hypothetical Ag lattices with the K atoms removed[4]. The observation suggested the stability of both alloys as derived from the interaction of Ag $5p$ orbitals filled by electrons donated by the K atoms. Since then, the Zintl-Klemm concept[5] has been invoked to elucidate the structure and chemical bonding of various binary intermetallic compounds at high pressures[6–9], such as the metal superhydrides[10] showing unprecedentedly high superconductivity critical temperatures. Despite the simplicity and apparent successes of the concept, the fundamental assumption of a large electron transfer between elements presenting large electronegativity differences has not been thoroughly scrutinized to establish a sound theoretical foundation to explain the experimental results.

With this study, we wish to understand better the crystalline structure and bonding of a prototypical series of binary alloys of K and Ag. This is accomplished by analyzing the crystalline structures and the electron density distribution obtained from experiments and comparing them to our theoretical calculations. For this purpose, we recorded and analyzed the X-ray diffraction patterns of $KAg_2$, $K_2Ag$ and $K_3Ag$ compounds synthesized from mixtures of K and Ag subjected to low pressures at room temperature.

Electron density maps were derived by the maximum entropy method (MEM)[11] using the intensity of the Bragg reflections and the Fourier synthesis obtained from Rietveld refined powder X-ray diffraction patterns[12]. In earlier studies[13–15], we have shown that the MEM analysis of powder X-ray diffraction patterns measured under high pressure in diamond anvil cells can effectively unveil how electron density distributions (ED) change under pressure. For instance, we observed the hybridization of Si $s$, $p$, and $d$ orbitals, leading to the metallization of densified silicon[13]. In this study, we aim to gain qualitative insights into how the electron density distribution changes, hence addressing how the chemical bonding evolves with the application of pressure in K-Ag alloys. Plane-wave density functional theory calculations were performed to support our analysis. The electron density distribution and wavefunction computed were analyzed by employing density-based and projected localized atomic basis methods to characterize the nature of electron interactions using conventional and the familiar bonding descriptors.

The key findings of this investigation are succinctly summarized as follows. K atoms donate their valence electrons to the system. The number of electrons shared by each K atom does not change substantially with pressure upon the formation of the respective K-Ag alloys, namely, $KAg_2$, $K_2Ag$, and $K_3Ag$. In the case of $KAg_2$, the valence electrons donated by K are utilized to form K-Ag and Ag-Ag covalent bonds. For $K_2Ag$, we reinterpreted the previous studies using our X-ray diffraction patterns and found a K and Ag disordered hexagonal structure, different from the previously suggested ordered structure. For $K_3Ag$, it is found that the K atoms occupied both the tetrahedral and octahedral voids of the face-centred cubic (FCC)

[1]Department of Physics and Engineering Physics, University of Saskatchewan, Saskatoon, SK, S7N 5E2, Canada. [2]Laboratoire de physique des solides denses, Department of Physics, University of Ottawa, Ottawa, ON, K1N 6N5, Canada. ✉e-mail: john.tse@usask.ca

 1

lattice formed by Ag atoms, strengthening the K-Ag bonds, while no Ag-Ag bonds are present. The electron density distribution derived from the experimental data indicates that the K-K interactions are weak yet discernible for all structures identified. Our observations and results diverge from the idealized complete electron transfer supported by the Zintl-Klemm model of alloy formation.

The outline of the paper is as follows. First, we provide details on the powder X-ray diffraction experiments and essential information regarding the Rietveld full-pattern refinements and the analysis by MEM. Then, we elaborate on the density functional calculations and the methodologies employed to characterize chemical bonding. Discussion on the structures and evolutionary pathways of the dense alloys. A comprehensive analysis of experimental and theoretical charge density distribution is provided to assess the chemical bonding. We conclude with a critical comparison between experimental and theoretical results.

## Methods

### Powder X-ray diffraction using synchrotron radiation

Samples of potassium (Strem Chemicals; ampouled, 99% min.) and silver (Sigma-Aldrich; powder, 99.99% min.) were prepared and loaded in gasketed diamond anvil cells ($P < 25$ GPa at 295 K) in appropriate different molar ratios in an inert (Ar) atmosphere presenting less than 1 ppm of oxygen and moisture. Most samples were constrained in Re gaskets. Tests indicated that T301 stainless steel gaskets were also appropriate. Once loaded without a pressure-transmitting medium, reactants were checked for purity by X-ray diffraction below the reactivity threshold pressure and, if adequate, were further compressed and decompressed in a series of X-ray diffraction experiments to study phase stability and changes. Pressures were measured at room temperature by the photoluminescence of $Al_2O_3$:$Cr^{3+}$ microspheres (about 10 μm in diameter), loaded with the samples. Typically, the spectral broadening of the photoluminescence developed at higher pressures, increasing uncertainty to about 0.6 GPa above 10 GPa.

X-ray diffraction images were collected at the Wiggler High Energy beamline at the Brockhouse Sector for X-ray Diffraction and Scattering of the Canadian Light Source using photons at λ = 0.3497 Å, following a proven experimental method[16]. Given the inhomogeneous nature of the microscopic samples, X-ray diffraction maps were recorded from each sample with a spatial resolution of about 20 μm. Hence, at a given sample location corresponding to a specific K/Ag ratio, X-ray diffraction images were recorded as a function of pressure. The images were processed for calibration and data reduction[17] and converted to X-ray diffraction patterns. In most instances, the X-ray diffraction patterns consisted of mixed crystalline phases of unreacted K and/or Ag and pressure-induced compounds. A preliminary analysis of the X-ray diffraction patterns was conducted using XRDA[18] to identify the alloy phases and provide initial cell parameters for further Rietveld analysis.

### Rietveld refinements and analysis by MEM

Phase identification and indexing of the X-ray diffraction patterns were further conducted using the cell indexing module in the General Structure Analysis System II (GSAS-II) software[19]. Initial background points computed using the automatic algorithm were fitted using a 20-term Chebyshev polynomial in GSAS-II. Additional points were added manually when necessary to achieve a more accurate description of the irregular backgrounds. The peak profile was described by a pseudo-Voigt function. 3D Fourier maps were calculated by the internal modules of GSAS-II.

The Le Bail method implemented in Jana2020[20] was employed to extract the structure factors needed for the MEM analysis, accomplished using Dysnomia[21]. MEM calculations were initiated from uniform densities and employed the zeroth-order single-pixel approximation (ZSPA)[21] as the optimization algorithm. Except for the analysis of $K_2Ag$, which will be addressed later, the convergence of the MEM calculation was achieved for all reported results.

## Computation details

All structural optimizations, charge densities, wavefunctions and electronic band structures were calculated using the Vienna ab initio Simulation (VASP) code[22] and Projector Augmented Plane Waves (PAW) potential[23] and Perdew-Burke-Ernzerhof (PBE) exchange-correlation functional[24]. The PAW potentials employed $3p^6 5s^1$ and $4d^{10} 5s^1$ as valence states for K and Ag, respectively. The k-point mesh used in the calculations were $21 \times 21 \times 10$, $7 \times 7 \times 10$, and $7 \times 7 \times 7$ for $KAg_2$, $K_2Ag$ and $K_3Ag$, respectively. The default energy cutoff of 249.8 eV for the PAW potentials was used. The Wannier90 code[25] interfaced with VASP was used to obtain the band structure from a GW[26] calculation and disentangle the electronic bands to generate the Wannier orbitals. One of the objectives of this study is to unveil the electron interactions of the alloys in the well-known chemical picture, addressing, for instance, atomic charges, orbital hybridization, and bond order. The electronic distributions can be analyzed with the charge density or the projection of localized basis sets for K and Ag atoms to the plane-wave wavefunctions. Bader's quantum theory of "atoms-in-molecules" (QTAIM)[27] was used for charge density analysis. The topology of electron density is the physical manifestation of the forces acting within the system. The electron density distribution is determined by the interaction between two nuclei and chemical bonding. The associated topological properties at the critical points, e.g., the bond critical point (BCP), provide a good measure of the strength of the interaction. Topological analysis was performed with the Critic2 program[28] using the charge density calculated by VASP. In addition, net atomic charge and bond order were calculated using the density-derived electrostatic and chemical (DDEC6) method implemented in the Chargemol program[29]. This new method partitions electron density into chemically meaningful components and delineates the electron density of the system into atomic contributions, providing insight into charge distribution, bonding patterns, and related properties. Natural bond orbital (NBO)[30,31] is a powerful method to reveal chemical bonding by deriving local orbitals (i.e., the Lewis structure) from the electron density of the system. However, to obtain a real space description of the bonding, it is necessary to project localized atomic orbitals onto the plane wave. The reliability of the projection is subject to the quality of the atomic basis sets and the parameters used in the computation of localized properties. In our calculations, the quadruple-zeta Gaussian atomic orbital (AO) basis sets, def2-QZVP[32] were used. Orbitals with very small exponents (diffuse) were removed. The quality of the NBO projection was accessed by ensuring the total spread after projection is less than $1.0 \times 10^{-2}$. The calculation of the natural atomic orbital (NAO) occupation reflects the hybridization of the atom.

## Results

### Analysis of X-ray diffraction patterns

X-ray diffraction patterns of crystalline $KAg_2$, $K_2Ag$, and $K_3Ag$[1,2] were observed from compression/decompression experiments starting from mixtures of elemental K and Ag. The X-ray diffraction patterns at 4.13 ($KAg_2$), 4.41 ($K_2Ag$), and 5.88 GPa ($K_3Ag$) are shown in Fig. 1. The X-ray diffraction patterns indicate the presence of residual K mixed from the respective K-Ag alloy phases. The K contamination, however, does not impede the quality of the patterns for Rietveld refinements and MEM analysis. The different phases identified are discussed in detail below.

### $KAg_2$

$KAg_2$ is the lowest pressure compound among the K-Ag alloys identified from the X-ray diffraction data. The phase is observed between 1.56 GPa and 4.2 GPa at room temperature. The results of our analysis confirmed the unit cell reported[2]. It has a hexagonal $P6_3/mmc$ space group with lattice parameters $a = 5.7237$ Å, $c = 9.7411$ Å and Z = 4 at 2.26 GPa. In the unit cell, the Ag atoms occupy the 2a and 6h sites and the K atoms are located at the 4f sites. The X-ray diffraction pattern (Fig. 2) shows the presence of cubic phase I of K ($Im\bar{3}m$, K-I). A Rietveld refinement on the combined patterns was successful with $R_w = 0.91\%$ with the $KAg_2$ to K-I ratio of about 10:1 (see Fig. 2).

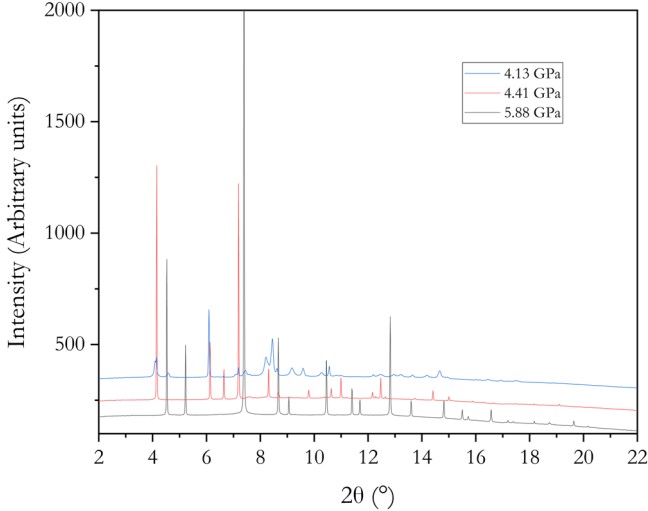

**Fig. 1 | Representative X-ray diffraction patterns for K-Ag alloys.** KAg₂ (top) K₂Ag (middle) and K₃Ag (bottom).

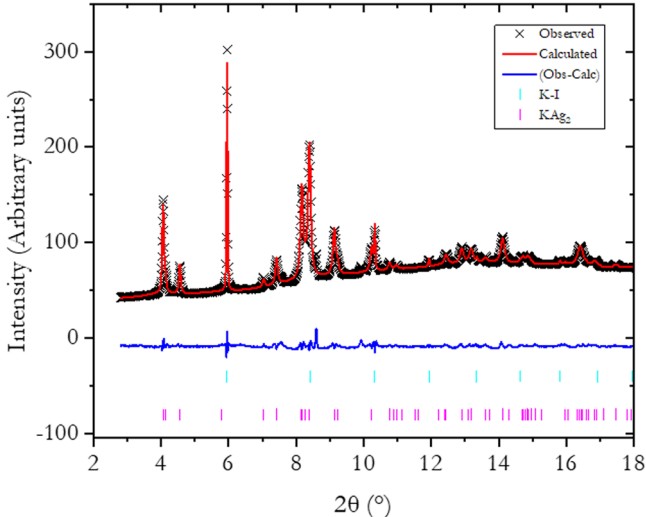

**Fig. 2 | Comparison of the Rietveld refined and measured X-ray diffraction pattern of KAg₂ at recorded 2.36 GPa.** The pure potassium contribution to the pattern belongs to the low-pressure body-centered cubic phase (Im$\bar{3}$m K-I). The KAg₂ was obtained by releasing the pressure from a higher-pressure phase. Inevitably, different phases in small quality are present in the sample, resulting, for instance, in a weak line at 2θ = 9.7°.

An important feature of KAg₂ (Fig. 3a) that was overlooked in an earlier study[2] is that its crystalline structure can be described by (Ag)₄ clusters constructed from two-dimensional (2-D) layers of corner-shared triangular (Ag)₃ units in the *ab* plane (plane A, Fig. 3b) linked through coplanar apical Ag atoms (plane B, Fig. 3c) in the *c*-direction. The in-plane Ag-Ag distances are identical at 2.84 Å. The distance from the apical Ag to the Ag in the plane is slightly longer at 2.92 Å. The K-Ag distances range from 3.36–3.49 Å. The shortest K-K separation is 3.39 Å.

When viewed down the *c*-axis, the A, B planes of Ag stacking resemble that of the precursor FCC Ag with the Ag atoms in the center of the hexagons removed (Fig. 3c). In KAg₂, these "voids" are replaced with pairs of K atoms above and below and alternate the positions between planes A and B.

A Le Bail refinement[20] of the X-ray diffraction pattern was performed to extract the Bragg reflection intensities. Using a set of 19 reflections, convergence in the MEM analysis[21] was reached. The electron density (ED) distribution obtained is consistent with the crystalline structure and the Fourier map resulting from the Rietveld refinement (Fig. 4). The MEM-derived valence electron topology (Fig. 4b) shows that the electron densities of the Ag atoms in the planar triangular clusters are spread along the *c*-direction. The electron density at the K atoms is also slightly distorted from the spherical distribution. The former observation suggests that there may be a covalent interaction between the Ag atoms. The electron density distribution revealed from the 3D Fourier maps, assuming spherical atomic scattering factors, also shows slight distortions around the K and Ag atoms (Fig. 4a). Indeed, an examination of the (004) plane of the Ag₃ clusters in both the MEM electron density distribution and the Fourier map indicates the accumulation of electrons between the Ag atoms (Fig. 4c, d, respectively).

## K₂Ag

The synthesis of K₂Ag and its crystalline structure was first reported in ref.1. At 4 GPa, the unit cell was reported to be the hexagonal *P6/mmm* space group with Z = 1 and lattice parameters *a* = 5.5434 Å and *c* = 3.770 Å. In the hexagonal unit cell, Ag atoms are at the 1*a* (0,0,0) position, whereas the K atoms are at the 2*d* (1/3,2/3,1/2) site. Both atomic species form layers stacked along the *c*-axis. In the original study, Atou et al.[1] noted the absence of (00 l) reflections in the X-ray diffraction pattern, which, with the assumption of preferred orientation, led to the conclusion of the *P6/mmm* space group.

In the present study, high-resolution X-ray diffraction patterns recorded at 2.24 and 3.22 GPa (upon pressure decrease) show the existence of the K₂Ag phase. Apart from a minor contribution of the K-I phase, the patterns would be seen in agreement with the proposed *P6/mmm* unit cell[1] if the (001) reflection could be ignored. However, other absent reflections, particularly at higher scattering angles, indicate this is not the case. Table 1 lists the predicted but unobserved reflections below 15° (2θ) with the *P6/mmm* space group[1].

**Fig. 3 | Structural motifs of crystalline KAg₂.** The crystal structure is constructed from the precursor face-centered cubic (FCC) Ag. **a** shows interconnected Ag-tetrahedra (open circles) and K atoms (solid, purple). The Ag atoms are arranged in (**b**) 2-D quadrilateral (plane B) and hexagonal (Kagome) Ag planes (plane A). **c** shows the Ag sites replaced by K.

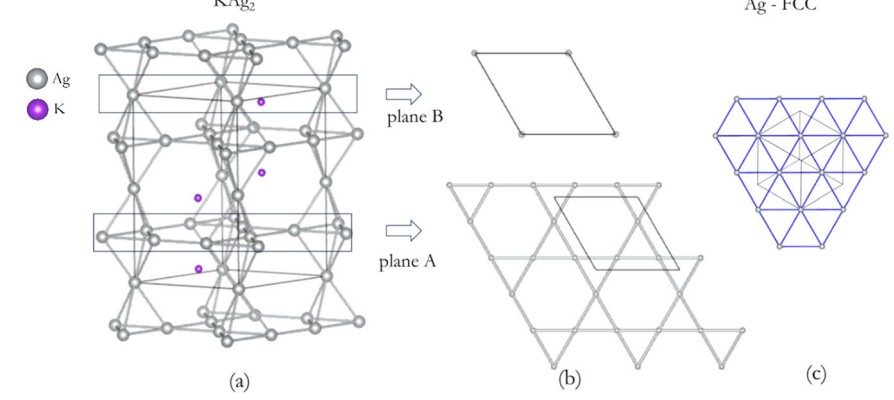

**Fig. 4 | Charge Density Distribution in KAg₂.**
Charge density distribution obtained (**a**) from
Fourier analysis of the Rietveld refined results using
spherical atomic scattering factors using an iso-
surface value of 13.0 e/$a_0^3$ and (**b**) from the MEM
analysis using an iso-surface value of 6.5 e/$a_0^3$. Elec-
tron density distribution of KAg₂ in the (004) plane
from (**c**) Fourier map and (**d**) MEM-derived. Note
the contour scales for the two plots are different. The
units of the scales are e/Å³.

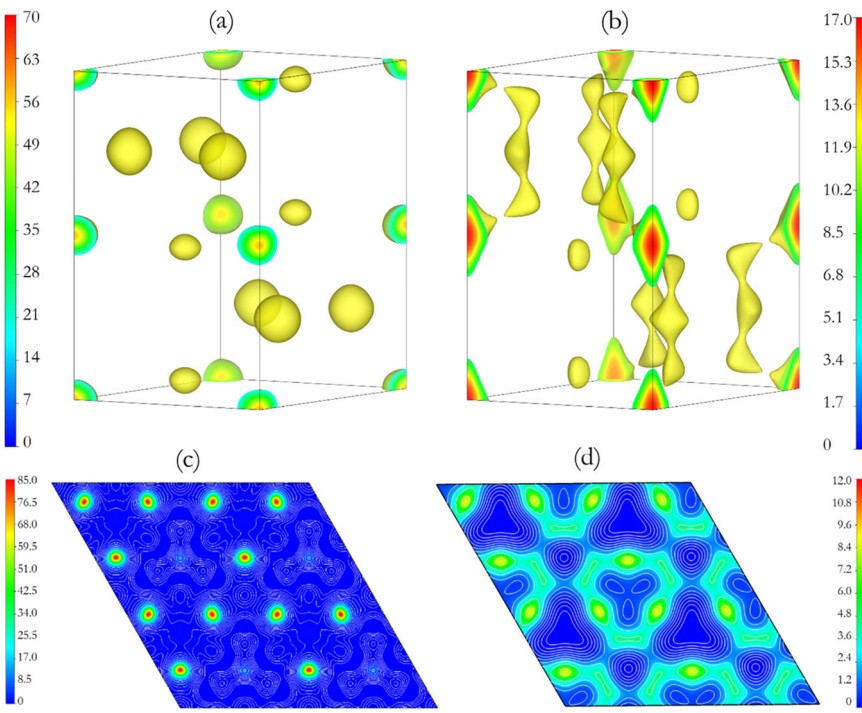

### Table 1 | Predicted but unobserved reflection for *P6/mmm* space group

| hkl | 2θ (°) |
|---|---|
| 0 0 1 | 5.19 |
| 2 1̄ 1 | 8.87 |
| 0 0 2ᵃ | 10.40 |
| 1 0 2ᵃ | 11.20 |
| 3 0 1 | 13.52 |
| 4 2̄ 1 | 15.33 |
| 0 0 3 | 15.63 |

ᵃindicates a reflection not reported in the earlier study[1] but observed in our experiments.

A unit cell indexing performed using GSAS-II and JANA2000 predicted a *P6₃/mmc* space group, which matches better the X-ray diffraction pattern with the same lattice parameters. The new space group assignment eliminates the discrepancies related to the unobserved reflections as given in Table 1 (except for (301) line observed at 13.52°) and other high-angle reflections. However, by assuming the *P6₃/mmc* space group, Ag atoms would need to occupy the 2*a* sites and K atoms the 4*f* sites. In this case, the resulting crystalline structure gives unrealistically short interatomic distances. There are two possible solutions to circumvent this issue to explain the observed X-ray diffraction patterns: (1) partial occupancy of the atomic sites or (2) to invoke the existence of a superlattice by doubling the cell along the *c*-axis.

Before further addressing the pattern refinement problem, we analyzed the electron density distribution derived from MEM analysis in both *P6/mmm* and *P6₃/mmc* space groups. No convergence was achieved using either space group. Nevertheless, upon examination of the crude EDs, we found hints of atomic disordering along the *c*-axis in both space groups, indicated by a continuous electron density distribution in the [001] direction. The results lead us to advance the possibility of a superlattice. We then constructed a superlattice unit cell by doubling the *c*-axis for both space groups. In both models, a very good Le Bail fit with $R_p$ less than 1% was obtained. However, the *P6/mmm* supercell predicts an additional peak at a low scattering angle that was not observed in the experiment.

As shown in Fig. 5, ED maps obtained in both space groups clearly show a continuous electron density between the Ag atoms along the *c*-axis. The "extended" charge distribution is too large for Ag-Ag bonding. We speculate that there may be a "disordered" structure with Ag partial occupancy halfway along the *c*-axis and between the two equivalent Ag atoms in Wyckoff position 2*b*. In addition, the electron density distribution around the K sites in the middle of the unit cell also appears to show disorder. Based on this observation, we propose four possibilities for the structure of K₂Ag: a supercell (i) with no partial occupancy (i.e., with all atoms in their respective Wyckoff symmetry positions) (ii) with the (0,0,1/4) site partially occupied with Ag (iii) with K atoms displaced from the ideal Wykoff sites, or (iv) with one K and all Ag sites allowed to be partially occupied. We performed a Rietveld refinement on the above-mentioned models without considering the preferred orientation. Table 2 reports the $R_w$-factor for the refined models in the *P6₃/mmc* space group.

The Rietveld refinements show gradual but significant improvements in $R_w$ when a supercell with doubling of the *c*-axis is adopted and disordering K and Ag atoms when partial occupancy is considered. The results lend support to the "continuous" electron distribution between the Ag atoms due to the disordering as revealed by MEM analysis. The best fit achieved is when partial occupancy of disordered K and Ag sites is considered. It is noteworthy to indicate that low-angle, high-intensity reflections bias the goodness-of-fit parameter, $R_w$, of the Rietveld refinement. However, a qualitative comparison of the calculated and experimental patterns supports the results of the Rietveld analysis. With no partial occupancy, some unobserved reflections appeared at high angles and their intensities did not fit well. In contrast, when partial occupancy is considered, those reflections disappear, and concomitantly, fits to the intensity of the high-angle reflections improves. Even with the best structural model, not all high-angle features are adequately accounted for. As mentioned in the earlier study, the discrepancy could be due to a preferential orientation of the sample. Indeed, we obtain a significant improvement using a preferential orientation model that reduces $R_w$ to 0.83%. Furthermore, the refined isotropic thermal parameters are also very reasonable. A comparison of the measured and fitted X-ray diffraction patterns is shown in Fig. 6. The final model gives a stoichiometry of K₂Ag₀.₉₄, almost identical to the ideal composition. The

**Fig. 5 | ED maps derived from MEM analysis.** Supercells in $P6_3/mmc$ (left) and $P6/mmm$ (right) space groups with non-distorted atomic positions were used. In both space groups, the Ag atoms are located at the corners and the edges of the unit cell, and the K atoms are inside the unit cells. The units of scale are $e/Å^3$.

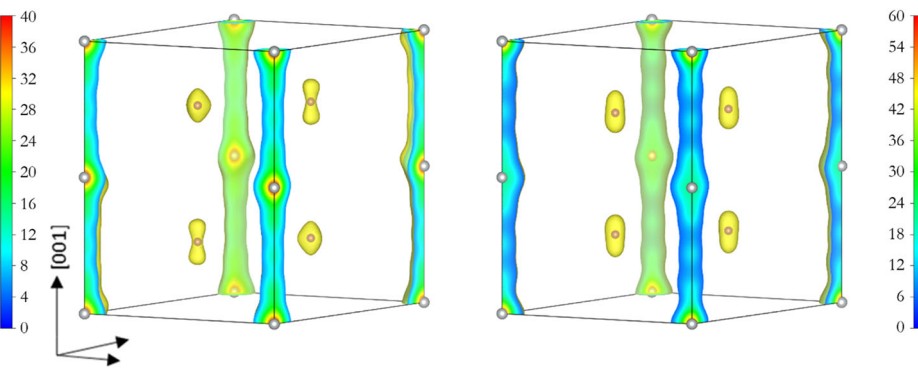

## Table 2 | $R_w$ agreement factor for the Rietveld refinement of the four proposed structural models in the $P6_3/mmc$ space group

| Structure | $R_w$ (%) |
|---|---|
| Supercell - no partial occupancy | 8.14 |
| Supercell - K partial occupancy | 6.76 |
| Supercell - Ag partial occupancy | 3.78 |
| Supercell- K and Ag partial occupancies | 2.62 |

Details of the refinement procedure is described in the K₂Ag section.

## Table 3 | Parameters obtained from Rietveld refinement of the data presented in Fig. 6, considering the disordered K₂Ag model in the $P6_3/mmc$ space group

| Name | x | y | z | frac | site | Uiso (Å²) |
|---|---|---|---|---|---|---|
| Ag-1 | 0 | 0 | 0 | 0.7243 | 2a | 0.1682 |
| Ag-2 | 0 | 0 | 0.25 | 0.2122 | 2b | 0.0531 |
| K-1 | 0.3333 | 0.6667 | 0.25 | 1 | 2c | 0.4931 |
| K-2 | 0.6667 | 0.3333 | 0.3866 | 0.5 | 4f | 0.3577 |

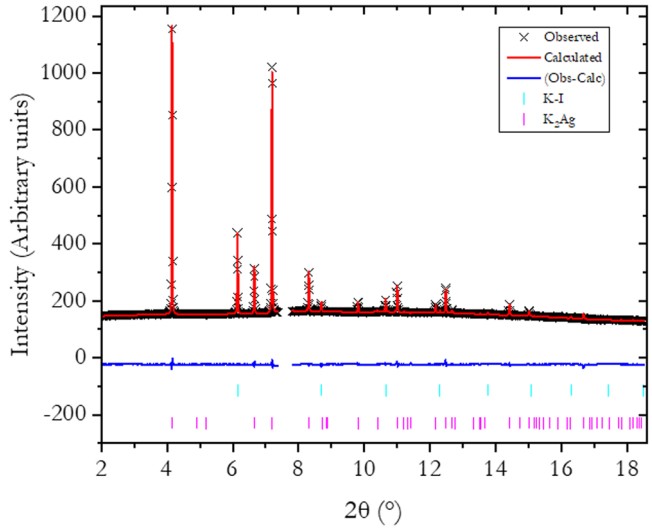

**Fig. 6 | Comparison of Rietveld refined and measured X-ray diffraction pattern of K₂Ag.** Refinement was performed in the $P6_3/mmc$ supercell with disordered Ag and K atoms (for details see K₂Ag section). For this refinement, a preferred orientation is considered leading to an $R_w = 0.83\%$.

structural parameters obtained from the Rietveld refinement of the final model are presented in Table 3.

A similar analysis was performed considering the $P6/mmm$ space group. In this case, the agreement is significantly inferior. Moreover, a diffraction peak at around 2.5°, predicted for the supercell, is not accounted for in the experimental X-ray diffraction pattern. In addition, many predicted high-angle reflections are also not present in the data, resulting in a poor fit. We rule out the possibility of a unit cell with the $P6/mmm$ space group on this basis. Our analysis thus indicates that the best structural model consistent with the measured diffraction patterns of K₂Ag has a $P6_3/mmc$ supercell, with $a = 5.604$ Å, $c = 7.757$ Å and $Z = 2$. The 3D Fourier map

(Fig. 7a) with ED obtained from the MEM analysis (Figs. 5 and 7b) shows a remarkable resemblance.

It is worth noting that although the Ag-Ag distance is longer than in the elemental phase, K₂Ag maintains the morphology of 2D honeycomb layers, which stack in a (disordered) …A-A-A… manner with layers intertwined with K atoms.

### K₃Ag

K₃Ag is the highest-pressure K-Ag phase observed in this study, up to 13.2 GPa. It typically appears at pressures above 5.5 GPa. It has a cubic $Fm\bar{3}m$ structure and the unit cell parameter decrease from $a = 7.89$ Å at 5.5 GPa to $a = 7.41$ Å at 13.2 GPa[1]. The X-ray diffraction patterns show the presence of a small amount of the K-I phase, which is taken into account during the full-pattern refinement without complications. Le Bail and Rietveld refinements were conducted on five diffraction patterns measured at different pressures. A comparison of the refined and measured pattern at 5.5 GPa is shown in Fig. 8. In this case, the goodness-of-fit factor, $R_w$ equals 0.84%. In the unit cell, the Ag atoms occupy the 4a site and the K atoms the 4b and 8c sites. The crystalline structure can simply be described with K atoms inserted into the octahedral (4b) and tetrahedral (8c) sites of the FCC Ag lattice and expanding the unit cell parameter from 4.042 Å of pure Ag at 5.5 GPa to 7.89 Å for the K₃Ag unit cell at the same pressure.

The MEM analysis converged readily on all X-ray diffraction patterns. The 3D ED maps derived from the Rietveld refined patterns at five pressure points are presented in Fig. 9. The 3D electron density distribution around the K and Ag atom locations shows no apparent deviation from a spherical distribution. A detailed examination of the charge density on the (100) and (110) planes, encompassing the Ag-Ag and Ag-K atoms in the unit cell, shows hints of K-K and K-Ag interactions as observed in Figs S1–S10. However, no discernible interactions between Ag atoms are found. This observation is at odds with a previous suggestion[4] based on theoretical band structures whereas Ag accepts electrons from K to populate its 5p orbitals leading to the formation of Ag-Ag bonds.

### Theoretical analysis

**Electronic structure.** Although there are ambiguities in the structure of K₂Ag, nonetheless, the models proposed above are all based on the same

**Fig. 7 | Comparison of 3D Fourier map and electron density distribution obtained from MEM of KAg₂. a** 3D Fourier map for the disordered structure of K₂Ag and (**b**) superposition of the rescaled Fourier map with the electron density distribution (grey colour) obtained from the MEM analysis.

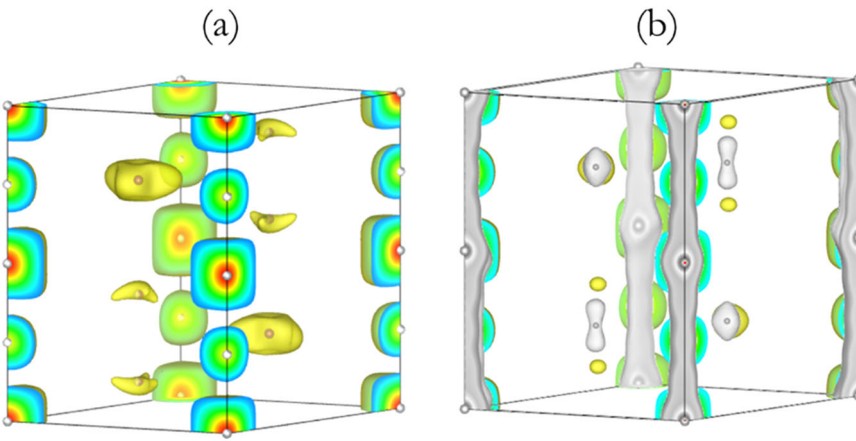

(a)     (b)

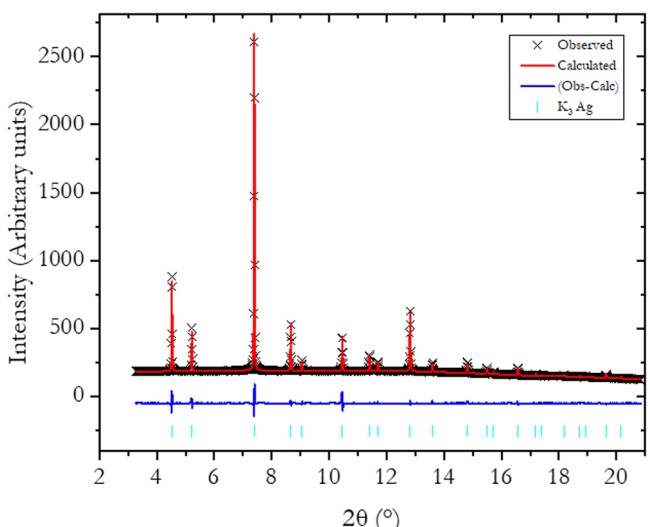

**Fig. 8 | Rietveld refined and measured X-ray diffraction pattern of K₃Ag.** The pattern (*Fm$\bar{3}$m*) was recorded at 5.5 GPa.

basic structural motif, i.e., the stacking of hexagonal 2D-Ag layers with K atoms situated between the layers, as derived from the *P6/mmm* space group. As it is not possible to compute the electronic structure of the *P6₃/mmc* disordered solid, we employed the ordered *P6/mmm* structure in the calculation for comparison. The GW disentangled band structures together with the dominant Wannier orbitals in the indicated electronic band regions for KAg₂, K₂Ag, and K₃Ag are depicted in Fig. 10. Examination of the electron band dispersions indicates substantially different chemical interactions for the three K-Ag alloys considered.

In KAg₂, the band structure shows mixing of the participating K and Ag atoms. The Ag 4*d* bands are not distinguishable and are heavily involved in the bonding. Disentanglement into K and Ag atoms is more complicated. The Ag 4*d* band is broad and extends from −5.5 eV to −4.0 eV. The shape of the dominant Ag 4*d* Wannier orbitals (Fig. 10, top) is distorted from the ideal atomic orbitals, indicating hybridization with other orbitals. K-Ag bonding is observed from −3 eV to the Fermi level. Above the Fermi level, K orbitals with "*d*" character start to appear. To accurately disentangle the bands in the lower valence state in K2Ag, we employed a projection-based method with the inner and outer disentanglement energy windows set to −6 eV and 0 eV, respectively. As depicted in Fig. 10 (middle), the disentangled Wannier orbital shows the band mixing is primarily due to the 4*d* and 5*s* orbitals of the Ag atom. The localized bands between −4 eV and

−5.2 eV correspond to 4*d* orbitals of the Ag atoms, which do not participate strongly to the bonding. Similarly, the bands around the Fermi level are disentangled by freezing the electron states up to the Fermi level, with the inner and outer energy windows set to −2 eV and 4 eV, respectively. The Wannier functions around the Fermi level are mostly for Ag 4*p*. The Wannier orbitals indicate that the Ag 4*d* and 5*s* electrons are strongly mixed with the K orbitals, showing a significant chemical bonding. In K₃Ag, the Ag atom dominates the lower energy valence states (Fig. 10 bottom). The 4*d* bands between −4.5 and 5.0 eV are very narrow and completely isolated from the 5*s* with a gap of 1 eV. Using inner and outer energy windows of −1 eV and 2 eV, respectively, with the frozen state set to the Fermi level (i.e. only states up to the Fermi level are included in the wannierization), the Wannier orbitals extracted from the upper valence state and lower conduction bands reveal a significant contribution from K atoms to the bonding states. The Wannier orbital shows a K *s*, *p*, and *d* ($d_{z^2}$ and $d_{x^2-y^2}$) hybrid orbital.

The analysis of the calculated projected electron density of states (PDOS) corroborates the qualitative description of the chemical bonding in the K-Ag alloys. The Ag and K projected density of states for the three alloys are shown in Fig. 11. With 8 Ag atoms (each with 10 *d*-orbitals) and four K atoms in the unit cell of KAg₂, the 4*d* density of states dominates the lower valence level from −7 to −3 eV. The 4*d* contribution relative to 5*s* and 5*p* is relatively significant up to the Fermi level (Fig. 11). In comparison, the PDOS of Ag in K₂Ag is more localized between −4.4 and 4.0 eV, albeit mixed with a small amount of 5*s*, and did not extend into the upper valence region. The dominance of Ag 5*s* between −4.0 to −0.8 eV is discernible. The PDOS of Ag is a mixture of 5*s* and 5*p* orbitals close to the Fermi level (Fig. 11). Clear separation of the 4*d* and 5*s* bands is displayed in K₃Ag (Fig. 11). Contributions from the 5*p* orbitals are becoming more significant around −0.8 eV.

The K PDOS in KAg₂ shown in Fig. 11 indicates strong participation of the 3*p* orbitals in the entire valence region. In particular, the high PDOS between −5.5 and −3.0 eV overlaps with the Ag 4*d* and 5*s* bands (Fig. 11), showing, unambiguously, Ag-K bonding. The PDOS of K profiles from −3.0 eV to the Fermi level exhibit dissimilarities to the PDOS of Ag (Fig. 11), indicating the likelihood of predominantly K-K interactions. In K₂Ag, the distribution of the PDOS of K (Fig. 11) of predominantly 3*s* is quite broad between −4.4 eV and −2.0 eV and corresponds well with the 4*d* and 5*s* bands of the Ag atom, indicating K (4*s*) – Ag (4*d*, 5*s*) bonding. The predominantly 4*s* PDOS profile near the Fermi level from −1.60 eV is not similar to the Ag, so it likely indicates K-K interactions. The K "*d*" character becomes stronger above the Fermi level. The PDOS of K in K₃Ag is less complicated (see Fig. 11). It overlaps with the Ag 5*s* from −4.2 to 2.5 eV (cf. Fig. 11). The peak at −0.8 eV matches the Ag 5*p* in the PDOS of Ag suggesting Ag-K bonding. It is noteworthy to highlight that the PDOS of the Ag atom near or at the Fermi level decreases as the K content increases in the alloy.

**Fig. 9 | MEM-derived ED of K₃Ag at the indicated pressures.** The ED at 5.50 GPa has an iso-surface value of 0.77 e/$a_0^3$. All other pressures have an iso-surface value of 2.0 e/$a_0^3$.

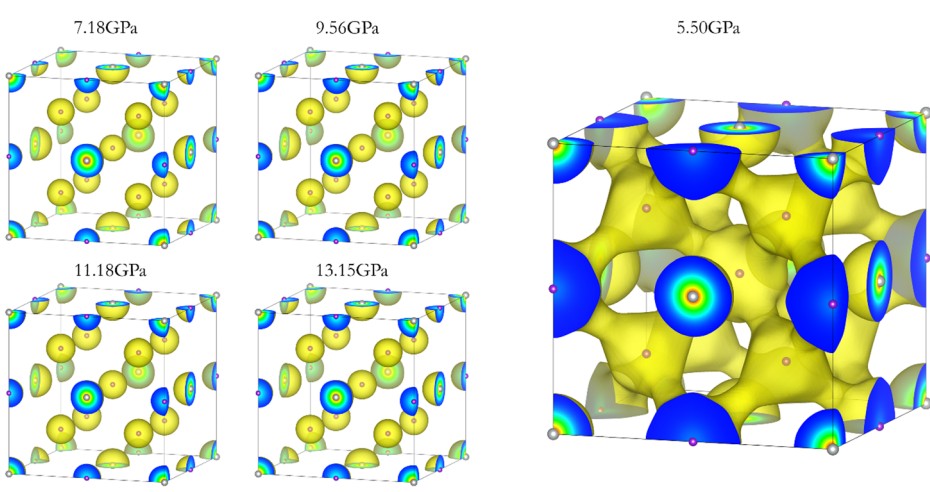

7.18GPa　　9.56GPa　　5.50GPa

11.18GPa　　13.15GPa

**Fig. 10 | GW electronic band structures and dominant Wannier orbitals.** (top) KAg₂, (middle) K₂Ag, and (bottom) K₃Ag.

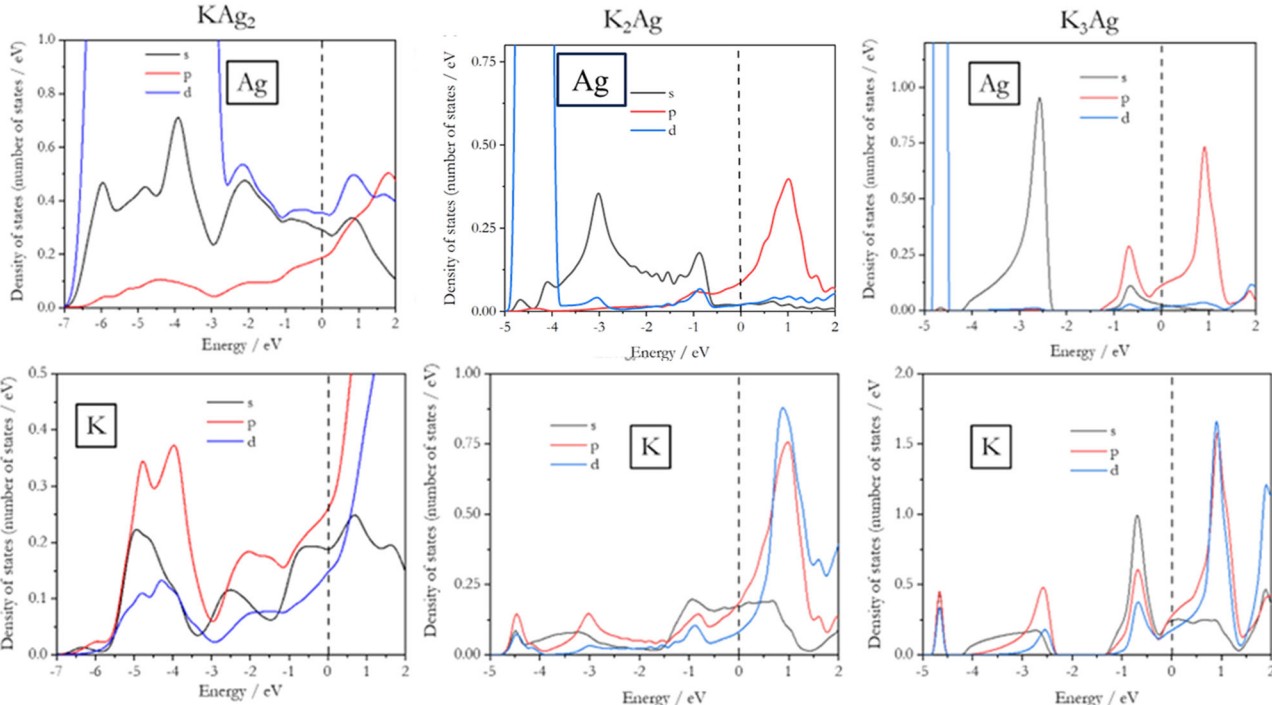

**Fig. 11 | Plane-wave projected electron density of states (PDOS).** Orbitally resolved Ag and K PDOS in the valence band of KAg$_2$, K$_2$Ag and K$_3$Ag. Note that different scales were used for the PDOS plots.

## Discussion

Examining the computed electronic structures provides an overall picture of a change in the nature of chemical bonding in KAg$_2$, K$_2$Ag, and K$_3$Ag. Particularly, a progressive localization of the Ag 4$d$ band is most noticeable. Atomic Ag has an electron configuration [Kr]4$d^{10}$5 $s^1$ and the $d$-shell is filled formally. However, the 5$s$ to 4$d$ orbital proximity facilitates the $sd$ hybridization. This is evident in the case of KAg$_2$. On increasing K content, the Ag 4$d$ orbitals become more localized and do not participate in the bonding. The anticipated charge transfer from the electropositive K to Ag certainly affects the valence bonding. The conventional method of charge partitioning implemented in most plane wave codes is subjected to arbitrariness when choosing the atomic volume. Therefore, unbiased methods of assigning atomic charges based on the calculated density are preferred. For this purpose, we computed the atomic charge based on QTAIM and the more recent DDEC6 algorithms. For comparison, we also evaluate the NBO orbital occupancy using quadrupole zeta (def-QZVP) atomic basis sets. Table 4 summarizes the computed atomic charges.

The atomic charges calculated from the plane wave electron densities (*viz.* QTAIM and DDEC6) are consistent with each other. As expected, the K atoms lose the 4 s electrons to the system. The number of electrons lost for each K atom seems to decrease with the increasing K content in a compound and the number of electrons gained by Ag increases. The NAO charges computed from the projection of localized atomic orbitals to the plane wave wavefunctions show a similar trend. The calculated natural orbital occupancies for Ag in KAg$_2$, K$_2$Ag, and K$_3$Ag are $4d^{10}5s^{1.20}5p^{0.07}$, $4d^{10}5s^{1.64}5p^{0.64}$ and $4d^{10}5s^{1.55}5p^{0.11}$, respectively; and for K, $4s^{0.15}4p^{0.13}3d^{0.02}$, $4s^{0.29}4p^{0.24}3d^{0.02}$ and $4s^{0.22}4p^{0.29}3d^{0.02}$, respectively. The results show K $sp$ and Ag $dsp$ hybridization. The NBO populations are consistent with the density analysis, which shows that the total number of electrons being transferred from K to Ag increases with K content, but the population of the Ag 5$p$ orbitals does not increase significantly. This observation indicates that although Ag is more electronegative, it does not fully accommodate all the available K valence electrons when the potassium concentration is high.

The bonding in the alloys can be characterized by the bond critical points (BCP) from the analysis of the electron density topology with QTAIM and the bond order, which can be used as an indicator for bond

## Table 4 | Computed atomic charges for K and Ag for the different K-Ag alloys

| Alloy | Atom | QTAIM | DDEC6 | NBO |
|---|---|---|---|---|
| KAg$_2$ | K | +0.666 | +0.639 | +0.715 |
| | Ag | −0.334, −0.348 | −0.295, −0.32 | −0.38, −0.36 |
| K$_2$Ag | K | +0.585 | +0.509 | +0.480 |
| | Ag | −1.171 | −1.018 | −0.961 |
| K$_3$Ag | K | +0.519, +0.506 | +0.411, +0.474 | +0.451, +0.430 |
| | Ag | −1.531 | −1.361 | −1.311 |

strength, as computed from the DDEC6 method. In all cases, the Morse sums are satisfied in the QTAIM analysis.

As illustrated in Fig. 12, many BCPs are found in KAg$_2$. All BCPs are located and situated almost halfway along the K-K, K-Ag, and Ag-Ag bond paths. Therefore, the QTAIM analysis clearly reveals that the interactions are found between all the atoms in KAg$_2$. Bond orders (BO) calculated by DDEC6 using the same charge density support this assignment (Table 5). Although not a genuine two-electron bond, the calculated Ag-Ag BO values of 0.39–0.47 are reasonably high. The planar triangular Ag-Ag BO of 0.47 is higher than that of the in-plane-apical Ag-Ag bond which is 0.39. This is consistent with the difference for the Ag-Ag bond lengths (Table 6). The Ag-K and K-K bond orders are comparably weaker, around 0.09 and 0.03, respectively. Thus, electrons from K atoms are not completely shared with Ag atoms to form anions, as described in the simple Zintl-Klemm picture. Instead, the electrons are involved to some extent in forming different chemical bonds.

Following the electron density topological analysis, three distinct types of bond critical points between Ag-Ag, Ag-K, and K-K atom pairs are found in K$_2$Ag (Fig. 13). The density ($\rho(r_{BCP})$), which can be estimated, is an indicator of bond strength. In this case $\rho(r_{BCP})$ for Ag-Ag, Ag-K and K-K are computed to be 0.015, 0.008, and 0.007 ($e/a_0^3$), respectively. These values can be correlated with the calculated bond order of 0.30, 0.1 and, 0.07, given in the same order.

**Fig. 12 | Bond critical points for KAg₂.** Ag-Ag, Ag-K, and K-K bond critical points (BCP) found by Critic2[28] for KAg₂.

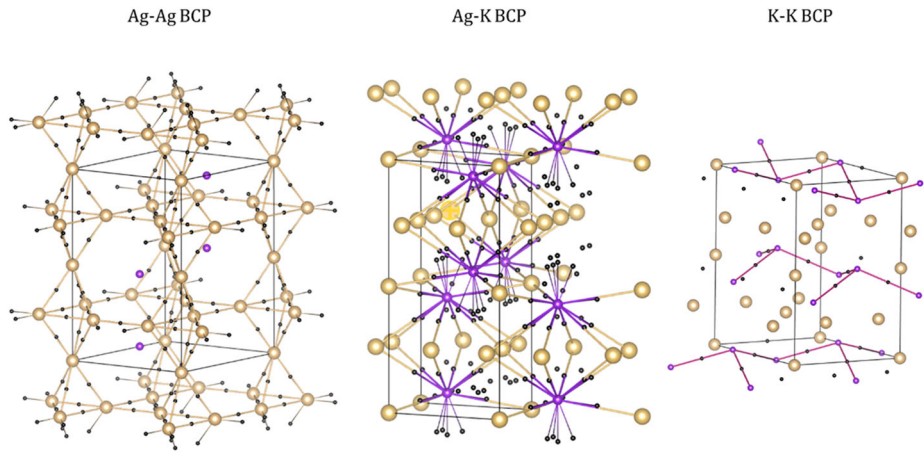

Ag-Ag BCP        Ag-K BCP        K-K BCP

**Table 5 | Bond order in KAg₂, K₂Ag and K₃Ag calculated by the DDEC6 algorithm[29]**

| Alloy | KAg₂ | K₂Ag | K₃Ag |
|---|---|---|---|
| Ag-Ag | 0.46 (planar) 0.39 (apical) | 0.30 | 0.0 |
| Ag-K | 0.09, 0.08 | 0.11 | 0.20, 0.07 |
| K-K | 0.03 | 0.07 | 0.08 |

**Table 6 | List of experimental atomic separations in KAg₂, K₂Ag and K₃Ag**

| Alloy | Pressure / GPa | Ag-Ag / Å | Ag-K / Å | K-Ag / Å |
|---|---|---|---|---|
| KAg₂ | 1.57 | 2.82, 2.93 | 3.39, 3.40 | 3.46, 3.84 |
| K₂Ag | 2.36 | 2.91 | 3.35 | 3.43, 3.84 |
| K₃Ag | 5.56 | 5.58 | 3.42, 3.95 | 3.42 |

**Fig. 13 | Bond critical points for K₂Ag.** The hexagonal $P6/mmm$ at 4 GPa was used in the calculation. Colour code: purple = K atom, silver = Ag atom and black = bond critical points (BCP).

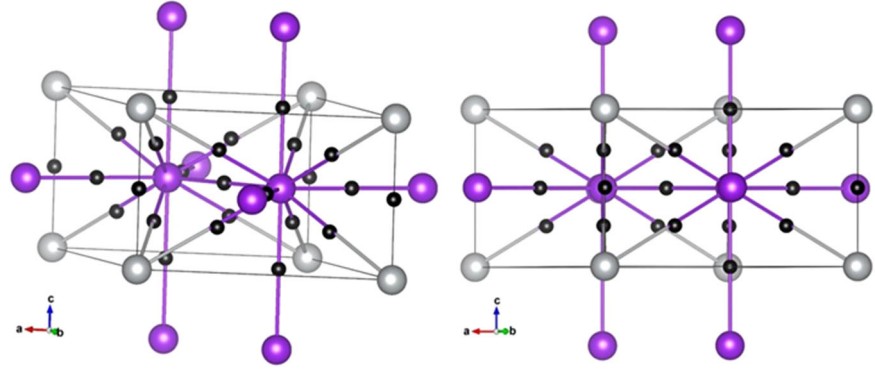

BCP and bond order analyses were performed on K₃Ag at 6.4 GPa. The primitive cell used in the calculation and the BCP are illustrated in Fig. 14. Ag-K and K-K BCPs are found, but no Ag-Ag BCP is found. Two crystallographically distinct K atoms are located in the tetrahedral and octahedral sites of the Ag FCC lattice. There are noticeable differences in the K-Ag interaction for the two sites. The $\rho(r_{BCP})$ of 0.012 $e/a_0^3$ at the tetrahedral site has a slightly smaller atomic charge +0.506$e$ is higher than the K at the octahedral site with $\rho(r_{BCP})$ of 0.008 $e/a_0^3$ and a charge of +0.519$e$. BO calculations confirmed this assessment. The calculated BO for K-Ag at the tetrahedral site is 0.195 and a much smaller BO of 0.070 at the octahedral site. QTAIM analysis also revealed K-K interactions with a relatively small $\rho(r_{BCP})$ of 0.008 $e/a_0^3$. The corresponding BO is 0.076.

In passing, it is informative to elucidate the chemical bonding in the K-Ag alloys in the viewpoint of the crystal structures. A tabulation of the Ag-Ag, Ag-K, and K-K separations in the crystal structure of KAg₂, K₂Ag ($P6/mmm$), and K₃Ag are summarized in Table 6. Interestingly, the Ag-Ag, Ag-K, and distances in the two lower pressure phases, KAg₂ at 1.57 GPa and K₂Ag at 2.36 GPa, are very similar, even after considering the pressure difference. So, it is not surprising that the atomic interactions revealed from the theoretical analysis, that is, the presence of Ag-Ag, Ag-K, and K-K

bonds, are also similar. Only in the higher-pressure K₃Ag are the bonds somewhat weakened and the ionicity of K and Ag becomes larger. The Ag-Ag separation increases abruptly from 2.91 Å in K₂Ag to 5.58 Å in K₃Ag with the pressure change of 3.2 GPa. The long distance between the Ag atoms prevents orbital overlap. The earlier postulation suggests that interactions between the Ag 5$p$ orbitals[4] are thus invalid. Instead, it is observed from our experimental results and theoretical analysis that the acceptance of the K electrons into the Ag 5$s$ and 5$p$ orbitals facilitates Ag-K bonding.

## Conclusion

The crystalline structures, electronic structure, and chemical bonding in K-Ag alloys formed under the application of pressure[1] have been studied by high-resolution powder X-ray diffraction using synchrotron radiation and the analysis of the electronic structures. We found a systematic structural trend with the insertion of K into the Ag FCC lattice. When K atoms are in contact with Ag atoms, they lose their valence electron and become smaller ions that diffuse progressively into the FCC Ag framework, forming the different alloy structures. We also have re-evaluated the structure of K₂Ag. Through full-pattern Rietveld refinements, we found that a disordered structure with a supercell of $P6_3/mmc$ symmetry fits better the observed

**Fig. 14 | Bond critical points of for cubic K₃Ag.** The calculations were performed in the primitive cell at 6.4 GPa.

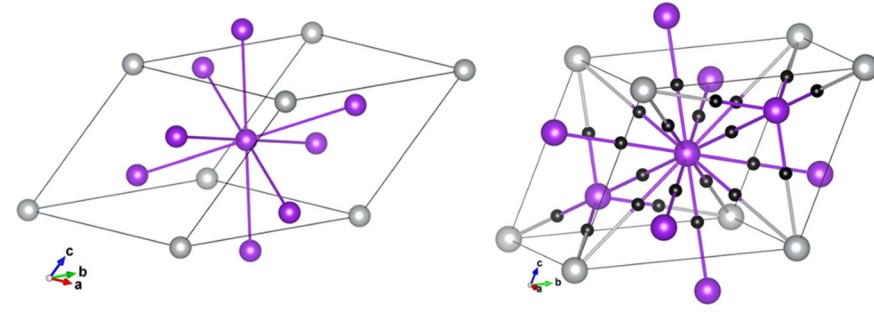

X-ray diffraction pattern. Electron density distribution maps obtained from MEM analyses indicated Ag-Ag and Ag-K bonding in the alloys. This observation is supported by the examination of the nature of disentangled Wannier orbitals, QTAIM topological analysis[27,28] of the charged density and the projection to localized atomic orbitals NBO[30] and DDEC6[29] analyses. All results indicate that interactions of K 4 *s*, 4*p*, and 3*d* with mix Ag 5 *s*, 5*p*, and 4*d* orbitals dominate the valence bands of KAg₂ and K₂Ag. In comparison, the Ag 4*d* orbitals are localized in K₃Ag. Succinctly, electron transfer from K to the 5 *s* and 5*p* orbitals of Ag atoms helps to maintain Ag-K bonding in these structures. Contrary to a previous suggestion[4], no Ag-Ag bonding *via* the overlap of Ag 5*p* orbitals is found. Nevertheless, the Zintl-Klemm charge-transfer concept[5] is broadly followed. This phenomenon is also observed in the hydrogen-rich superhydrides, as demonstrated in the study on the bonding of strontium hydrides[10]. However, a straightforward application of this concept to elucidate high-pressure structures of inter-metallic compounds[6] should be exercised with caution.

## Data availability

Rietveld analysis of diffraction patterns was performed with the open-source codes, JANA2020 and GSAS. The Dysnomia code was used for Maximum Entropy Analysis. All electronic structure calculations were performed with a licensed VASP code. Wannier functions were computed with Wannier90. Any relevant data are available from the authors upon reasonable request.

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

## Acknowledgements

The experimental work described in this paper was performed at the Canadian Light Source Inc., a national research facility at the University of Saskatchewan, which is supported by the Canada Foundation for Innovation (CFI), the Natural Sciences and Engineering Research Council (NSERC), the National Research Council of Canada (NRC), the Canadian Institutes of Health Research (CIHR), the Government of Saskatchewan, and the University of Saskatchewan. JST acknowledges support from the Digital Alliance of Canada for the allocation of computing resources. DR thanks University of Saskatchewan, the Canadian Light Source for an INSPIRE fellowship. JST and SD are supported by the Natural Sciences and Engineering Research Council of Canada through the Discovery Grant Program.

## Author contributions

J.S.T. and S.D. designed the study. S.D. prepared all samples, conducted the experiments, and initially processed the data. D.R. performed the Rietveld refinements and the MEM analysis and interpreted the experimental data. N.U., H.K. and J.S.T. performed and analyzed the calculations. N.U., D.R., H.K., S.D., and J.S.T. interpreted the results. J.S.T. drafted the manuscript. N.U., D.R., H.K., S.D., and J.S.T. revised the manuscript. N.U. and D.R. contributed equally to the study.

## Competing interests

The authors declare no competing interests.
