## [Peer Review File · Communications Chemistry]

Reviewers' comments:

Reviewer #1 (Remarks to the Author):

The manuscript performs a comprehensive analysis of the evolution of the crystalline structure and chemical bonding of a prototypical series of binary high-pressure K-Ag alloys using high-resolution powder X-ray diffraction and theoretical calculations. It emphasizes that the Zintl-Klemm concept should be exercised with caution in elucidating high-pressure structures of intermetallic compounds. Whereas the methodology used is justified and the research content is interesting to some degree, the analysis of the electronic structure underlying it is subpar. As a consequence, the authors are asked to address the following before recommending the manuscript for publication.

1. The authors repeatedly mention the Zintl-Klemm concept and emphasize it should be exercised with caution in elucidating high-pressure structures of intermetallic compounds. However, the definition of the Zintl-Klemm concept is missing. Please provide a more specific explanation in the manuscript.
2. The experimental method used in this study has been verified in earlier studies. Is there any work that has validated the use of electron density distribution and wavefunction as an indicator for bonding features? Please add representative literature to the Introduction.
3. In Figure 3, the structural/atomistic presentation of the model systems is currently well done but a legend with the type of atoms should be added to help the reader understand intuitively and easily the concept of the model.
4. The picture arrangement in the manuscript is too scattered, which is not conducive to readers' reading and analysis. It is recommended to improve the quality and reorganize the figures, for example, merging Figures 4, 5, and 6 describing the KAg₂ electronic structure into Figure 4.
5. Mentioned in the section of K₃Ag "it is mentioned that a detailed examination of the charge density on the (100) and (110) planes, encompassing the Ag-Ag and Ag-K atoms in the unit cell, shows hints of K-Ag interactions". Can authors directly display the 2D electron density distribution on the (100) and (110) planes at five pressure points in Figure 10?
6. The authors should put the analysis of the projected electron density of states (PDOS) before the band structures and the dominant Wannier orbitals, to clearly explain the corresponding relationship between the energy of the localized band and Wannier orbitals. Redoing Figure 11 to ensure that all the Wannier orbitals are the same size, and supplementing the corresponding relationship between the Wannier orbital and the band structures of KAg₂.
7. It is recommended that the projected density of states of K and Ag in the same structure should be placed together and labeled as K_s, K_p, K_d, Ag_s, Ag_p, and Ag_d, respectively, and then put the projected density of states of KAg₂, K₂Ag, and K₃Ag in a horizontal arrangement in the same picture for comparative analysis.
8. The Conclusion in the manuscript should be concise and highlight the topic.

9. There are many methods to directly verify the chemical bonding between atoms in theoretical calculations, such as electron localization function(ELF) and crystal orbital Hamilton population (COHP). If necessary for subsequent research, the authors can perform relevant calculations.

Reviewer #2 (Remarks to the Author):

In this manuscript, the crystalline and electronic structure of K-Ag alloys prepared under high pressure (namely KAg₂, K₂Ag, and K₃Ag) has been thoroughly studied experimentally and computationally. The crystal structure of K₂Ag has been re-evaluated, and the previous suggestion of Ag-Ag bonding in K₃Ag has not been confirmed. Overall, the manuscript is clear and well-written, and the conclusions are sound and well-supported by experimental and computational evidence. A few comments/suggestions intended to clarify some minor points are as follows.

Check the axes labels in Fig. 1 with respect to the other figures depicting x-ray diffraction patterns.

Consider including an electron density color scale in Fig. 4 and similar figures.

Please include Uiso units in Table 3.

The x-ray diffraction pattern in Fig. 2 exhibits some unaccounted weak Bragg peaks, particularly around 9.8 and 11 degrees 2theta. An attempt at identifying these Bragg peaks would be very welcome.

The charge density distributions in Fig. 4a and 4b seem qualitatively different. Could Authors clarify the difference? Is the electron density scale in both figures the same?

In line 66, check the phrase "For K₂Ag, we reanalyzed using our x-ray diffraction patterns..."

Check the Angstrom symbol at line 220

Check duplicated phrases in lines 439-443

In line 524, "This link...". The phrase is vague. Please clarify.

Reviewer #3 (Remarks to the Author):

This manuscript revisits a series of high-pressure K-Ag phases to provide improved structural and electronic descriptions, using both experimental and computational methods. A disordered structure for KAg_2 , with partial occupancy in both K and Ag sites, is found to best fit the available experimental data. This is consistent with the observation of a “continuous” electron distribution between Ag sites in MEM analysis of the structure. In terms of the electronic structure, the Ag 4d orbitals are seen to become increasingly localized with K content. Bonding interactions are found between pairs of Ag atoms most prominently in KAg_2 , which features a network of Ag_4 tetrahedra, but also in an ordered model of K_2Ag , while the Ag-Ag distances in K_3Ag are too long to support Ag-Ag bonds. The electronic structure picture of these compounds is fleshed out beyond a simplistic accounting via the Zintl-Klemm concept of large electron transfers. The authors emphasize that careful tracking of electron transfer – which is not always complete – is needed to understand the electronic structure of phases at high pressure. They use a series of complementary analyses from methods aimed at discerning orbital hybridization to methods for charge calculation to trace the electronic trends as K content is increased.

There were a number of places where the clarity of the manuscript could be improved, both in the presentation of the data in figures as well as in the textual descriptions, where there were some confusing portions.

Computational details should be expanded (e.g. to include energy cutoffs and k-grids). Additionally, spilling values or other criteria to assess the quality of projection into the NBO basis and the resulting charges calculated would be useful information.

Line 20: “....contributing to K-Ag and Ag-Ag bonds in KAg_2 and KAg_2” Should be K_2Ag and KAg_2?

The legend for Figure 1 should include the nominal phases (KAg_2 , K_2Ag , and K_3Ag) as well as the pressure for clarity. In fact, since XRD patterns are provided for the three phases later in the text, this figure might not be necessary, or could be moved to a supplementary file?

The caption of Figure 3 is somewhat confusing. Part (b) states that plane A is rectangular and plane B triangular arrangements for Ag atoms, but in the figure image the plane B corresponds to simple hexagonal packing and plane A to a kagome layer. The rectangular description is confusing. In Part (c) it might also be helpful to show the KAg_2 structure from the same (overhead) perspective to emphasize the similarity or to rework panel (b) to show this more clearly.

In Figure 4 the unit cells are mirror images of each other, making it a little awkward to compare visually. An isovalue would also be helpful.

Does the appearance of large off-atomic peaks in the MEM CDD have any additional meaning? The “blobs” aren’t quite along the Ag-Ag internuclear axis. It appears that some prior MEM results exhibiting non-nuclear maxima (NMM) have been found to be artifacts of the MEM procedure. Is there evidence that these features are “real”? It’s a little strange to see that result next to the fairly standard Fourier map in panel (a). In the Fourier map, the Ag atoms with the largest dispersion are indeed the ones with the strange features in the MEM result, but the dispersion looks relatively isotropic.

For Figure 5, it looks like the atomic positions aren’t in the same places between the two images, almost like (a) is taken from a larger supercell?

In Figure 6, it would be nice to see not just the MEM but also the basic atomic positions in the two space group models.

What experimental evidence could confirm the superstructure solution?

Are the partial occupancies in the final refined structure linked in any way (e.g. if the “middle”/“extra” Ag position is occupied is there more likely to be a K vacancy nearby?)

Line 340 “A detailed examination of the charge density on the (100) and (110) planes, encompassing the Ag-Ag and Ag-K interactions in the unit cell, shows hints of K-Ag interactions” Could these charge density plots be provided, possibly in Figure 10, which might not need all five separate densities to be shown?

Theoretical analysis

In Figure 11, please label the band structures with the phases as well. Site-projected bands could be useful for understanding which are primarily derived from K and which from Ag. The middle panel appears to have the bands colored either blue or red; is this part of the Wannier analysis?

Lines 373-374 discussing the 4s and 5d orbitals of the Ag atom – is this meant to be the 4d and 5s orbitals of the Ag atom, and then again Ag 4d orbitals at the end of line 374?

Lines 378-379 “The Wannier orbitals indicate that the Ag 4d and 5s electrons are strongly mixed with the K orbitals, showing a significant chemical bonding” If this refers to K2Ag, could this be expanded upon?

Figures 12 and 13 could probably be combined into one figure, with say the top row being the Ag projected and the bottom row the K projected DOS in order for easier comparison. Figure 12b is also identical to Figure 13c, so it is difficult to understand some of the text, which must be referring to an image that is not present.

Lines 414-415 “...PDOS of the Ag atom near or at the Fermi level decreases as the K content increases in the alloy” This would be easier to see if the PDOS were plotted using the same scale, or the values could be provided in a table or in the text.

In Table 4, there looks to be a typo in the QTAIM results for K in K2Ag – if correct, this would be a very charge-unbalanced system. The K charges in K3Ag should also be labeled with their multiplicity.

For all of the charge decomposition strategies, it would be interesting to see e.g. charge spilling values which should help to capture the quality of the calculated charges.

Line 439 “K atoms lose the 3s electrons to the system” 4s? or 3d?

Lines 439-443 A sentence is repeated

The calculated natural orbital occupancy for Ag in K2Ag has a markedly higher 5p occupation than the rest. Does this reflect anything in particular?

The K orbitals are listed as 3s, 3p, 3d – should this be 4s/4p/3d?

Were the $\rho(r_{BCP})$ values calculated for KAg2 as well? This phase has the strongest Ag-Ag bonding according to the DDEC6 algorithm, it would be interesting to compare values.

Line 506 “Only in the higher-pressure K2Ag...” K3Ag?

A broader question: as pressure increases, K is expected to undergo an s->d transition and to behave somewhat like a transition metal rather than an alkali metal. Does this have any effect on the electronic structure of these phases which occur at progressively higher pressures?

Department of Physics
116 Science Place
University of Saskatchewan
Saskatoon, SK, S7N 5B2
Canada

Dear Reviewers,

Thank you for communicating your remarks on our manuscript titled “Structure and Chemical Bonding in High-Pressure K-Ag Alloys” by Nnanna Ukoji et al., with manuscript number COMMSCHEM-24-0169, and for the careful reading of the manuscript. We have revised it in accordance with your suggestions. We have addressed each comment in detail and made appropriate modifications to the manuscript (marked in blue). We hope the responses satisfy your concerns.

Best regards,

On behalf of all authors,

John S. Tse, Ph.D, D.Sc, FRS Canada

University Distinguished Professor

University of Saskatchewan Centennial Enhancement Chair

Canada Research Chair in Materials Science (2004-2018)

Reviewer #1 (Remarks to the Author):

The manuscript performs a comprehensive analysis of the evolution of the crystalline structure and chemical bonding of a prototypical series of binary high-pressure K-Ag alloys using high-resolution powder X-ray diffraction and theoretical calculations. It emphasizes that the Zintl-Klemm concept should be exercised with caution in elucidating high-pressure structures of intermetallic compounds. Whereas the methodology used is justified and the research content is interesting to some degree, the analysis of the electronic structure underlying it is subpar. As a consequence, the authors are asked to address the following before recommending the manuscript for publication.

1. The authors repeatedly mention the Zintl-Klemm concept and emphasize it should be exercised with caution in elucidating high-pressure structures of intermetallic compounds. However, the definition of the Zintl-Klemm concept is missing. Please provide a more specific explanation in the manuscript.

Response:

The Zintl-Klemm concept is a well-known empirical principle explaining the occurrence of binary intermetallic compounds, most commonly between alkali, alkaline-earth metals and groups 13, 14 and 15 elements. The principle combines the electron-counting rule and the electronegativity difference between the constituent elements. In an earlier study, this principle was invoked to elucidate the formation and structure of K-Ag alloys, using this principle at high pressure.

We now include a brief explanation in the manuscript. For a detail discussion of the underlying principle, we have also cited a new reference (ref. 5),

R. Nasper, *Z. Anorg. Allg. Chem.* 2014, 640, (14), 2639–2648

2. The experimental method used in this study has been verified in earlier studies. Is there any work that has validated the use of electron density distribution and wavefunction as an indicator for bonding features? Please add representative literature to the Introduction.

Response:

Indeed, several successful applications have been made using the maximum entropy method to analyze the electron distribution under ambient and high pressure. We have now cited the following articles:

J. Zhao, J. Reid, T. Iida, K. Takarabe, M. Wu, and J.S. Tse, Valence Charge Density of Multi-doped Mg₂Si Thermoelectric Materials from Maximum Entropy Method Analysis, *J. Alloys Compounds*, 681, 66, 2016.

R. Flacau, S. Desgreniers and J.S. Tse, Electron Density Topology of Cubic Structure I Xe Clathrate Hydrate at High Pressure, *J. Chem. Phys.*, 129, 244507, 2008.

J.S. Tse, R. Flacau, S. Desgreniers and J.Z. Jiang, Electron density topology of high-pressure Ba₈Si₄₆ from a Rietveld and maximum-entropy analysis, *Phys. Rev. B*, 76, 174109, 2007.

R. Flacau, C. I. Ratcliffe, S. Desgreniers, Y. Yao, D. D. Klug, P. Pallister, I. L. Moudrakovskia and J. A. Ripmeester, Structure and dynamics of ammonium borohydride, *Chem. Commun.*, 2010, 46, 9164-9166.

3. In Figure 3, the structural/atomistic presentation of the model systems is currently well done but a legend with the type of atoms should be added to help the reader understand intuitively and easily the concept of the model.

Response:

We thank the reviewer for this suggestion. A new legend indicating the type of atoms has now been added to Fig.3.

4. The picture arrangement in the manuscript is too scattered, which is not conducive to readers' reading and analysis. It is recommended to improve the quality and reorganize the figures, for example, merging Figures 4, 5, and 6 describing the KAg₂ electronic structure into Figure 4.

Response:

We thank the reviewer for this suggestion. We now have grouped the charge density of KAg_2 shown in the original Figs. 4 and 5 to a updated Fig. 4.

Fig. 6 in the original manuscript depicted the electron density of K_2Ag and is different from KAg_2 . We feel it is more appropriate to display it separately and retain the figure as the new Fig. 5.

5. Mentioned in the section of K_3Ag “it is mentioned that a detailed examination of the charge density on the (100) and (110) planes, encompassing the Ag-Ag and Ag-K atoms in the unit cell, shows hints of K-Ag interactions”. Can authors directly display the 2D electron density distribution on the (100) and (110) planes at five pressure points in Figure 10?

Response:

We thank the reviewer for this suggestion. To support the discussion in the text, we created a figure showing the 2D electron density in the [100] and [110] planes and included them as supplementary materials. Moreover, we included a 3D contour plot showing the charge density in the K_3Ag phase at 5.50 GPa depicting K-K interactions in the updated Fig. 9.

6. The authors should put the analysis of the projected electron density of states (PDOS) before the band structures and the dominant Wannier orbitals, to clearly explain the corresponding relationship between the energy of the localized band and Wannier orbitals. Redoing Figure 11 to ensure that all the Wannier orbitals are the same size, and supplementing the corresponding relationship between the Wannier orbital and the band structures of KAg_2 .

Response:

We feel the Wannier orbitals clearly illustrate the nature of the chemical interactions of the alloys. Therefore, we prefer to retain the order of discussion in the manuscript.

We have resized the figures as suggested.

7. It is recommended that the projected density of states of K and Ag in the same structure should be placed together and labeled as K_s , K_p , K_d , Ag_s , Ag_p , and Ag_d , respectively, and then

put the projected density of states of KAg_2 , K_2Ag , and K_3Ag in a horizontal arrangement in the same picture for comparative analysis.

Response:

To better compare the projected density of states of K and Ag of the same compound, the original Figs. 12 and 13 have been combined into a new Fig. 11, and in each case, the atomic contributions are clearly identified.

8. The Conclusion in the manuscript should be concise and highlight the topic.

Response:

We rewrote the Conclusion section. It is now more concise and highlights the major points: (1) a systematic structural trend of inserting K into the FCC Ag forming the various K-Ag alloys; (2) A new structure of K_2Ag is proposed; (3) Both experimental charge density and analyses of theoretical results indicate K-Ag and K-K interactions but no Ag-Ag bonding in the alloys.

9. There are many methods to directly verify the chemical bonding between atoms in theoretical calculations, such as electron localization function (ELF) and crystal orbital Hamilton population (COHP). If necessary for subsequent research, the authors can perform relevant calculations.

Response:

We thank the reviewer for the suggestions.

From our experience, the ELF values for metallic transition metal compounds are usually small and often less than 0.5. This makes the identification of “electron pairing” (localization) somewhat arbitrary. Therefore, we did not employ the ELF analysis.

We have performed COHP analysis on the K-Ag alloys with the LOBSTER code. In the version we used, the default atomic basis set for K unfortunately does not include diffuse orbitals to describe the valence s and p orbitals. Moreover, no d -type orbitals are available for the K-atoms. Thus, it is not possible to assess the anticipated K $s \rightarrow d$ hybridization.

To illustrate this point, the following two figures compare the projected DOS for K atoms in K_2Ag computed by LOBSTER and VASP codes. The features near the Fermi level are very different. The VASP calculations show significant K d -contributions not seen in the LOBSTER calculation, as the latter do not include the d -orbitals.

Surprisingly, the limitation of the default basis sets in LOBSTER, that may not be suitable to describe the bonding in high pressure structures, was not often discussed in the literature. Therefore, one must be cautious when extracting bonding information using the COHP and COOP methods implemented in the LOBSTER code.

Reviewer #2 (Remarks to the Author):

In this manuscript, the crystalline and electronic structure of K-Ag alloys prepared under high pressure (namely KAg_2 , K_2Ag , and K_3Ag) has been thoroughly studied experimentally and computationally. The crystal structure of K_2Ag has been re-evaluated, and the previous suggestion of Ag-Ag bonding in K_3Ag has not been confirmed. Overall, the manuscript is clear and well-written, and the conclusions are sound and well-supported by experimental and

computational evidence. A few comments/suggestions intended to clarify some minor points are as follows.

1. Check the axes labels in Fig. 1 with respect to the other figures depicting x-ray diffraction patterns.

Response:

The x and y axes labels, 2θ and intensity (arbitrary units), respectively, are now consistently applied to all the X-ray diffraction patterns.

2. Consider including an electron density color scale in Fig. 4 and similar figures.

Please include Uiso units in Table 3.

Response:

We thank the reviewer for this suggestion. We have now included the electron density color scale in Fig. 4 and similar figures as requested.

The colors of the electron density shown inside the iso-surface appear only at the slices of the atoms at the edges of the unit cell. They represent the electron distribution bounded by the iso-surface. It is important to note that the MEM analysis does not contain accurate information on core electrons due to the lack of high-angle data, and thus the coloring scheme is only qualitative.

We have now included the Uiso units (\AA^2) in Table 3 as requested.

3. The x-ray diffraction pattern in Fig. 2 exhibits some unaccounted weak Bragg peaks, particularly around 9.8 and 11 degrees 2θ . An attempt at identifying these Bragg peaks would be very welcome.

Response:

We thank the reviewer for the careful examination of the X-ray diffraction pattern. Indeed, there are two “impurity” peaks that we were not able to characterize. The KAg_2 pattern was obtained

by releasing the pressure from a higher-pressure phase, and we have found that different phases are always present in the sample. However, the intensities of these phases decrease with pressure, and the KAg_2 pattern that we report is the best in terms of minimizing extra diffraction lines. We now explain the extra peaks in the X-ray diffraction pattern in the caption of figure 2.

4. The charge density distributions in Fig. 4a and 4b seem qualitatively different. Could Authors clarify the difference? Is the electron density scale in both figures the same?

Response:

Fig. 4a was generated using MEM and Fig. 4b was derived from the Fourier analysis of the Rietveld refinement. So, the intensity scales are different and are now shown with the plots.

5. In line 66, check the phrase "For K_2Ag , we reanalyzed using our x-ray diffraction patterns..."

Check the Angstrom symbol at line 220

Check duplicated phrases in lines 439-443

Response:

We reworded the phrase "For K_2Ag , we reanalyzed using our x-ray diffraction patterns..." to now read "For K_2Ag , we reinterpreted the previous study using our X-ray diffraction pattern ..."

We indeed used the incorrect symbols for Angstrom. This is now corrected in the revised manuscript.

We removed the duplicated sentence in lines 439-443.

6. In line 524, "This link...". The phrase is vague. Please clarify.

Response:

We replace "This link" to "The observed trend"

Reviewer #3 (Remarks to the Author):

This manuscript revisits a series of high-pressure K-Ag phases to provide improved structural and electronic descriptions, using both experimental and computational methods. A disordered structure for KAg_2 , with partial occupancy in both K and Ag sites, is found to best fit the available experimental data. This is consistent with the observation of a “continuous” electron distribution between Ag sites in MEM analysis of the structure. In terms of the electronic structure, the Ag 4d orbitals are seen to become increasingly localized with K content. Bonding interactions are found between pairs of Ag atoms most prominently in KAg_2 , which features a network of Ag_4 tetrahedra, but also in an ordered model of K_2Ag , while the Ag-Ag distances in K_3Ag are too long to support Ag-Ag bonds. The electronic structure picture of these compounds is fleshed out beyond a simplistic accounting via the Zintl-Klemm concept of large electron transfers. The authors emphasize that careful tracking of electron transfer – which is not always complete – is needed to understand the electronic structure of phases at high pressure. They use a series of complementary analyses from methods aimed at discerning orbital hybridization to methods for charge calculation to trace the electronic trends as K content is increased.

There were a number of places where the clarity of the manuscript could be improved, both in the presentation of the data in figures as well as in the textual descriptions, where there were some confusing portions.

1. Computational details should be expanded (e.g. to include energy cutoffs and k-grids). Additionally, spilling values or other criteria to assess the quality of projection into the NBO basis and the resulting charges calculated would be useful information.

Response:

The k-point mesh used in the calculations were $21 \times 21 \times 10$, $7 \times 7 \times 10$ and $7 \times 7 \times 7$ for KAg_2 , K_2Ag and K_3Ag , respectively. The default energy cutoff of 249.8eV for the PAW potentials was used. The quality of the NBO projection was assessed by ensuring the total spread after projection is less than 1.0×10^{-2} . The NBO charges obtained are provided in Table 4. We added this clarification in the “Computational details” section.

2. Line 20: "... contributing to K-Ag and Ag-Ag bonds in KAg₂ and KAg₂..." Should be K₂Ag and KAg₂....?

Response:

Thank you to the reviewer for pointing out the typo. We have corrected the statement in the manuscript,

"... contributing to K-Ag and Ag-Ag bonds in KAg₂ and K₂Ag...."

3. The legend for Figure 1 should include the nominal phases (KAg₂, K₂Ag, and K₃Ag) as well as the pressure for clarity. In fact, since XRD patterns are provided for the three phases later in the text, this figure might not be necessary, or could be moved to a supplementary file?

Response:

The pressures corresponding to the relevant K-Ag phases were already indicated in the original figure.

To illustrate the progression of structure changes, we feel it is advantageous to compare the X-rays diffraction patterns of the three distinct phases of K-Ag studied here. We believe Figure 1 adds relevant information to the discussion.

4. The caption of Figure 3 is somewhat confusing. Part (b) states that plane A is rectangular and plane B triangular arrangements for Ag atoms, but in the figure image the plane B corresponds to simple hexagonal packing and plane A to a kagome layer. The rectangular description is confusing. In Part (c) it might also be helpful to show the KAg₂ structure from the same (overhead) perspective to emphasize the similarity or to rework panel (b) to show this more clearly.

Response:

In our opinion, Fig 3 a, b and c clearly illustrate the main structural features of the KAg₂ structure. In Fig. 3a, the periodic stackings of A and B layers are clearly depicted. Fig. 3b shows the atomic arrangement in the quadrilateral shape layer A and hexagonal (Kagome) layer B. We

mistakenly labelled layer A as “rectangular” in the original manuscript. The description in the caption has been changed. Fig. 3c provides another perspective of viewing the atom positions down the c -axes so no change to the figure was made.

5. In Figure 4 the unit cells are mirror images of each other, making it a little awkward to compare visually. An iso-value would also be helpful.

Response:

We thank the reviewer for suggesting the correction. The reason the figures appear as mirror images is due to the different perspectives of the plots. We have now replotted the figures from the same viewing direction to facilitate easier visual comparison. Additionally, we have included the iso-value as requested.

6. Does the appearance of large off-atomic peaks in the MEM CDD have any additional meaning? The “blobs” aren’t quite along the Ag-Ag internuclear axis. It appears that some prior MEM results exhibiting non-nuclear maxima (NMM) have been found to be artifacts of the MEM procedure. Is there evidence that these features are “real”? It’s a little strange to see that result next to the fairly standard Fourier map in panel (a). In the Fourier map, the Ag atoms with the largest dispersion are indeed the ones with the strange features in the MEM result, but the dispersion looks relatively isotropic.

Response:

We have performed MEM analysis with two prior densities, uniform charge density and charge density constructed from pro-atom distribution. Identical results were found. The Fourier synthesis assumes spherical atoms and attributes to the difference in the charge density distribution plot.

7. For Figure 5, it looks like the atomic positions aren’t in the same places between the two images, almost like (a) is taken from a larger supercell?

Response:

We apologize for the confusion. The unit cells of KAg_2 plotted in the original Fig. 5 were indeed different. Inadvertently, we displayed the cell along a and b axis from -1 to +2 in Fig. 5a but from -1 to +1 in Fig. 5b. Thus, the two figures are not the same. As shown below, we corrected this mistake and replotted both with the same dimension. To emphasize the similarity, we highlighted the Ag clusters with triangles (not shown in the revised Fig. 4 in the manuscript)

8. In Figure 6, it would be nice to see not just the MEM but also the basic atomic positions in the two space group models.

Response:

We thank the reviewer for the suggestion. We found it is too congested if we include the atom labels in the unit cell. Therefore, we identified the atoms in the caption of the new Fig. 5.

In both space groups, the Ag atoms are located along the edges of the unit cell and the K atoms are inside the unit cell.

9. What experimental evidence could confirm the superstructure solution?

Response:

As explained very clearly in the manuscript, we proposed five plausible structural models to interpret the X-rays diffraction pattern observed. The first model is the structure proposed by Atou et al. with ordered K and Ag. We ruled out this model with space group $P6/mmm$ because there are many predicted but not observed Bragg reflection in our higher resolution X-ray diffraction pattern. The other four models assume a supercell with a doubling of the c -axis and, with different degrees of K and Ag disorder. Based on our Rietveld refinements, we found that the model that reproduces the observed pattern best is a supercell with disordered K and Ag and a slight preferred orientation. This leads to the most probable structure according to the available data. Of course, it would always be ideal to examine the structure of K_2Ag using single-crystal diffraction. Producing suitable crystals of K_2Ag phase at high pressure, however, is beyond our capability at this moment.

10. Are the partial occupancies in the final refined structure linked in any way (e.g. if the “middle”/” extra” Ag position is occupied is there more likely to be a K vacancy nearby?)

Response:

We judged that this possibility is very low. Since the X-ray scattering factors of K and Ag are significantly different, exchanging K and Ag positions will strongly affect the calculated relative intensities. The result is expected to be grossly inconsistent with the observed pattern. Moreover, the MEM analysis clearly shows that the proposed Ag positions lead to a larger electron density than the K positions.

11. Line 340 “A detailed examination of the charge density on the (100) and (110) planes, encompassing the Ag-Ag and Ag-K interactions in the unit cell, shows hints of K-Ag interactions” Could these charge density plots be provided, possibly in Figure 10, which might not need all five separate densities to be shown?

Response:

We thank the reviewer for the comment. To illustrate some of the interactions, we have revised the original Figure 10 to include a 3D contour plot of the K_3Ag phase at 5.50 GPa (fig. 9).

Additionally, we have provided all the 2D plots of the (100) and (110) planes in supplementary materials as all these plots cannot be accommodated in single figures.

12. Theoretical analysis

In Figure 11, please label the band structures with the phases as well. Site-projected bands could be useful for understanding which are primarily derived from K and which from Ag. The middle panel appears to have the bands colored either blue or red; is this part of the Wannier analysis?

Response:

We thank the reviewer for this comment. We now labelled the corresponding K-Ag alloys to their band structures

As per the reviewer's follow-up question regarding the middle panel, the colored bands are indeed the disentangled Wannier bands of each orbital they represent.

13. Lines 373-374 discussing the 4s and 5d orbitals of the Ag atom – is this meant to be the 4d and 5s orbitals of the Ag atom, and then again Ag 4d orbitals at the end of line 374?

Response:

We thank the reviewer for catching this typo. The orbitals have been changed to 4d and 5s of Ag in the main text.

14. Lines 378-379 “The Wannier orbitals indicate that the Ag 4d and 5s electrons are strongly mixed with the K orbitals, showing a significant chemical bonding” If this refers to K₂Ag, could this be expanded upon?

Response:

Yes, the description refers to K₂Ag.

In the revised manuscript, we have added the following sentences to expand on the discussion.

The highly dispersive electronic band from -4.4 eV to the Fermi level exemplifies the strong K-Ag bonding. This explanation is further supported by the spread of the Ag and K projected

density of states over this energy region (vide supra) shown in the new fig. 12b and 12d.

15. Figures 12 and 13 could probably be combined into one figure, with say the top row being the Ag projected and the bottom row the K projected DOS in order for easier comparison. Figure 12b is also identical to Figure 13c, so it is difficult to understand some of the text, which must be referring to an image that is not present.

Response:

Thank you for the suggestion. We have now combined original Fig. 12 and 13 to a new Figure 11.

16. Lines 414-415 “.... PDOS of the Ag atom near or at the Fermi level decreases as the K content increases in the alloy” This would be easier to see if the PDOS were plotted using the same scale, or the values could be provided in a table or in the text.

Response:

Since the PDOS values of different alloys are quite different. In each case, it is necessary to plot the PDOS in different scales to reveal the details. We now added a note in the caption mentioning the different scales employed.

17. In Table 4, there looks to be a typo in the QTAIM results for K in K₂Ag – if correct, this would be a very charge-unbalanced system. The K charges in K₃Ag should also be labeled with their multiplicity.

Response:

We thank the reviewer for catching this typo. The charges for K₂Ag have been corrected in the revised manuscript.

18. For all of the charge decomposition strategies, it would be interesting to see e.g. charge spilling values which should help to capture the quality of the calculated charges.

Response:

The total spillover for occupied and occupied bands is less than 10^{-2} . This criterion is now reported in the manuscript.

19. Line 439 “K atoms lose the 3s electrons to the system” 4s? or 3d?

Response:

We thank the reviewer for identifying this typo. The correct K orbital is the 4s orbital and this has been updated in the revised manuscript.

20. Lines 439-443 A sentence is repeated

Response:

We thank the reviewer for pointing out the mistake. We have deleted the duplicated sentence.

21. The calculated natural orbital occupancy for Ag in K_2Ag has a markedly higher 5p occupation than the rest. Does this reflect anything in particular?

Response:

No, it does not. The high 5p occupation in K_2Ag is mainly due to charge transfer from K and Ag. This is not an indication of chemical interactions between the Ag atoms through the *p* orbitals. The calculation of the bond order by the DDEC6 method and analysis of the PDOS gave a more accurate description of the bonding.

22. The K orbitals are listed as 3s, 3p, 3d – should this be 4s/4p/3d?

Response:

We thank the reviewer for the question. We have made a mistake. The orbitals for the K atom are 4s, 3p, 3d and this have been updated in the revised manuscript.

23. Were the $\rho(r_{\text{BCP}})$ values calculated for KAg₂ as well? This phase has the strongest Ag-Ag bonding according to the DDEC6 algorithm, it would be interesting to compare values.

Response:

The calculated density at the bond critical point (BCP) for Ag-Ag is 0.032 -0.037 e/a_0^3 and 0.01 e/a_0^3 for K=Ag and 0.004 e/a_0^3 for K-K bonds. These values are comparable to the partial ionic interaction, such as in Li-H, (0.04 e/a_0^3) (ref: Bader, R.F.W. and Essen, H. (1984) The characterizations of atomic interactions. J. of Chem. Phys., 80, 1943–1960).

24. Line 506 “Only in the higher-pressure K₂Ag...” K₃Ag?

Response:

We thank the reviewer for pointing out this typo, it has been corrected in the revised manuscript.

25. A broader question: as pressure increases, K is expected to undergo an s→d transition and to behave somewhat like a transition metal rather than an alkali metal. Does this have any effect on the electronic structure of these phases which occur at progressively higher pressures?

Response:

In elemental K, the $s \rightarrow d$ transition occurs around 30 GPa, much higher pressure than the K-Ag alloys discussed here. One may speculate that when K-Ag is compressed to higher pressures, the $s \rightarrow d$ transition becomes more dominant and enhance K-K and K-Ag interactions forming new crystal structures.

REVIEWERS' COMMENTS:

Reviewer #1 (Remarks to the Author):

The authors have properly addressed all my concerns and thus I think it can be accepted as the current form.

Reviewer #2 (Remarks to the Author):

All comments and issues pointed out in my previous assessment of the manuscript were adequately dealt with by the authors. The revised manuscript is suitable for publication in Communications Chemistry.

Reviewer #3 (Remarks to the Author):

The manuscript describes a combined experimental/theoretical study of high-pressure K-Ag intermetallics to clarify the structural and electronic properties of three phases. The modifications made to the manuscript as a response to the first round of reviews have greatly improved the readability.

Line 221: "...the electron densities of the Ag atoms in the planar triangular clusters are spread along the c-direction. The electron density at the K atoms is also slightly distorted from the spherical distribution. The latter observation suggests that there may be a covalent interaction between the Ag atoms". Should this read instead "The former observation" referring to the deformation of the electron density of the Ag atoms?

There appears to still be a slight issue in figure 3, where plane A (in the caption, referring to the quadrilateral) and plane B (in the caption, the kagome net) are reversed. The caption has been updated but in the figure itself the quadrilateral layer is labeled as plane B and the kagome as plane A.

In the top panel of Figure 10, the Wannier functions from K and Ag atoms should be labeled as such. Similarly, the source of the red and blue band coloration (confirmed to be disentangled Wannier bands) in the band structure of K₂Ag should be mentioned in the text since the band structures for KAg₂ and K₃Ag are purely in black.

In Figure 11, the image provided for the Ag pDOS in K₂Ag (middle top) is identical to the image provided for the K pDOS in K₃Ag (bottom right). From the text, it appears that this is the correct image for the K pDOS in K₃Ag, but the actual Ag pDOS for K₂Ag then needs to be substituted in.

Dear Reviewer,

Thank you for communicating your remarks on our manuscript titled “Structure and Chemical Bonding in High-Pressure K-Ag Alloys” by Nnanna Ukoji et.al. with manuscript number COMMSCHEM-24-0169. We appreciate the careful reading of our manuscript and your constructive feedback. We have revised the manuscript in accordance with the suggestions provided. Below are our responses to the comments raised. We hope that our revisions satisfy the concerns raised and that the manuscript is now acceptable for publication.

Best regards,

John S. Tse, Ph.D, D.Sc, FRS Canada

University Distinguished Professor

University of Saskatchewan Centennial Enhancement Chair

Canada Research Chair in Materials Science (2004-2018)

Reviewer #3 (Remarks to the Author):

1. Line 221: "...the electron densities of the Ag atoms in the planar triangular clusters are spread along the c-direction. The electron density at the K atoms is also slightly distorted from the spherical distribution. The latter observation suggests that there may be a covalent interaction between the Ag atoms". Should this read instead "The former observation" referring to the deformation of the electron density of the Ag atoms?

Response:

We thank the reviewer for the correction. We have replaced the phrase as suggested. The sentence now reads: "...the electron densities of the Ag atoms in the planar triangular clusters are spread along the c-direction. The electron density at the K atoms is also slightly distorted from the spherical distribution. The former observation suggests that there may be a covalent interaction between the Ag atoms."

2. There appears to still be a slight issue in figure 3, where plane A (in the caption, referring to the quadrilateral) and plane B (in the caption, the kagome net) are reversed. The caption has been updated but in the figure itself the quadrilateral layer is labeled as plane B and the kagome as plane A.

Response:

We thank the reviewer for the correction. We have switched A and B in the figure description to match the labeling.

3. In the top panel of Figure 10, the Wannier functions from K and Ag atoms should be labeled as such. Similarly, the source of the red and blue band coloration (confirmed to be disentangled Wannier bands) in the band structure of K2Ag should be mentioned in the text since the band structures for KAg2 and K3Ag are purely in black.

Response:

It was a mistake in the band structure plot of K2Ag. We should have plotted everything in black. We have corrected the band structure of K2Ag, and now all band structures are illustrated as black lines.

4. In Figure 11, the image provided for the Ag pDOS in K2Ag (middle top) is identical to the image provided for the K pDOS in K3Ag (bottom right). From the text, it appears that this is the correct image for the K pDOS in K3Ag, but the actual Ag pDOS for K2Ag then needs to be substituted in.

Response:

We thank the reviewer for pointing out the discrepancy. We erroneously plotted the incorrect DOS for Ag in K2Ag. This has been corrected, and the new figure is illustrated below.